# A novel true random number generator based on a stochastic diffusive memristor

Hao Jiang[1], Daniel Belkin[1,2], Sergey E. Savel'ev [3], Siyan Lin[1], Zhongrui Wang[1], Yunning Li[1], Saumil Joshi[1], Rivu Midya[1], Can Li [1], Mingyi Rao[1], Mark Barnell[4], Qing Wu[4], J. Joshua Yang [1] & Qiangfei Xia[1]

The intrinsic variability of switching behavior in memristors has been a major obstacle to their adoption as the next generation of universal memory. On the other hand, this natural stochasticity can be valuable for hardware security applications. Here we propose and demonstrate a novel true random number generator utilizing the stochastic delay time of threshold switching in a $Ag:SiO_2$ diffusive memristor, which exhibits evident advantages in scalability, circuit complexity, and power consumption. The random bits generated by the diffusive memristor true random number generator pass all 15 NIST randomness tests without any post-processing, a first for memristive-switching true random number generators. Based on nanoparticle dynamic simulation and analytical estimates, we attribute the stochasticity in delay time to the probabilistic process by which Ag particles detach from a Ag reservoir. This work paves the way for memristors in hardware security applications for the era of the Internet of Things.

[1] Department of Electrical and Computer Engineering, University of Massachusetts, Amherst, MA 01003, USA. [2] Swarthmore College, Swarthmore, PA 19081, USA. [3] Department of Physics, Loughborough University, Loughborough LE11 3TU, UK. [4] Air Force Research Lab, Information Directorate, Rome, NY 13441, USA. Hao Jiang, Daniel Belkin and Sergey E. Savel'ev contributed equally to this work. Correspondence and requests for materials should be addressed to J.J.Y. (email: jjyang@umass.edu) or to Q.X. (email: qxia@umass.edu)

The internet of things (IoT) is a network of devices, sensors, and other items of various functionalities that interact and exchange data electronically[1]. Because of the explosive growth in the number of IoT objects (estimated to be 50 billion by 2020[2]) and overwhelming reliance on cyberspace, the existing hardware infrastructure is increasingly vulnerable to a wide range of security threats[3]. When software-based data securing methods are no longer sufficient because they are easily attacked, hardware security systems become critical. A true random number generator (TRNG) is a hardware component that generates a string of random bits, which can be used as a cryptographic key. It relies on intrinsic stochasticity in physical variables as a source of randomness. For example, thermal noise is often exploited by TRNGs via oscillator jitter[4], resistor-amplifier-Analog/Digital converter chains[5], or metastable elements with capacitive feedback[6]. Other approaches include using telegraph noise[7], current fluctuation in oxide after soft breakdown[8], or time-dependent oxide breakdown process[9]. However, all prior TRNGs have suffered from drawbacks in scalability, circuit complexity, or relied on post-processing such as a "Von Neumann corrector" to remove bias from the generated bit sequences[4–13].

Memristors[14–16], or resistive switching devices, have been proposed and demonstrated for a broad spectrum of applications[16–19] because of their attractive properties, such as low power consumption[20], fast switching speed[21], high endurance[22], excellent scalability[23], and CMOS-compatibility[24]. The intrinsic variation in switching parameters is a major challenge for some applications such as non-volatile memory[25]. However, this random behavior can be helpful in stochastic computing and hardware security applications[26–28]. For example, Huang et al. proposed a TRNG based on random telegraph noise (RTN) from the low resistance state of a W/TiN/TiON/SiO₂/Si memristor. The resulting circuit, however, proved difficult to activate and control because the probabilities of "0" and "1" were heavily

dependent on the applied voltages[29]. Balatti et al. demonstrated a TRNG using cycle-to-cycle and device-to-device voltage variations from Cu/AlO$_x$ and Ti/HfO$_x$ based memristors, respectively[30, 31]. Besides the need for complicated probability tracking and careful tuning of the applied voltage/current, a pair of SET and RESET pulses were required to generate each random bit since those memristive devices are non-volatile. More importantly, none of the aforementioned memristor based TRNGs passed all the 15 NIST Special Publication 800-22 randomness tests[32] even with post-processing of data, leaving the claimed true nature of the randomness debatable. Most recently, Wei et al. demonstrated a TRNG using randomness from small read-current fluctuation at certain resistance states in TaO$_x$-based devices. Sophisticated algorithms and circuits were needed to ensure the quality of generated binary bits before they could pass the NIST tests[33].

Here, we propose and demonstrate a novel TRNG based on a diffusive memristor, a newly developed volatile device that relies on the diffusion dynamics of metal atoms in the memristive layer[34, 35]. The device switches to a low-resistance state under a voltage pulse after a random delay time, and relaxes back to the high-resistance state spontaneously upon removal of the applied electrical bias. We use the intrinsic stochasticity of the delay time as the source of randomness to build a TRNG unit that consists of only a diffusive memristor, a comparator, an AND-gate, and a counter. Compared with previous TRNGs based on non-volatile memristors[30, 31], the self-OFF-switching behavior in the diffusive memristor greatly reduces the energy consumption since no RESET process is required. Our TRNG also has evident advantages in circuit complexity because the randomness is generated and harvested directly using simple elements. The diffusive memristor TRNG can easily be incorporated into memory subsystems, greatly increasing the security, and the area efficiency[33]. More importantly, the bits generated by our diffusive memristor

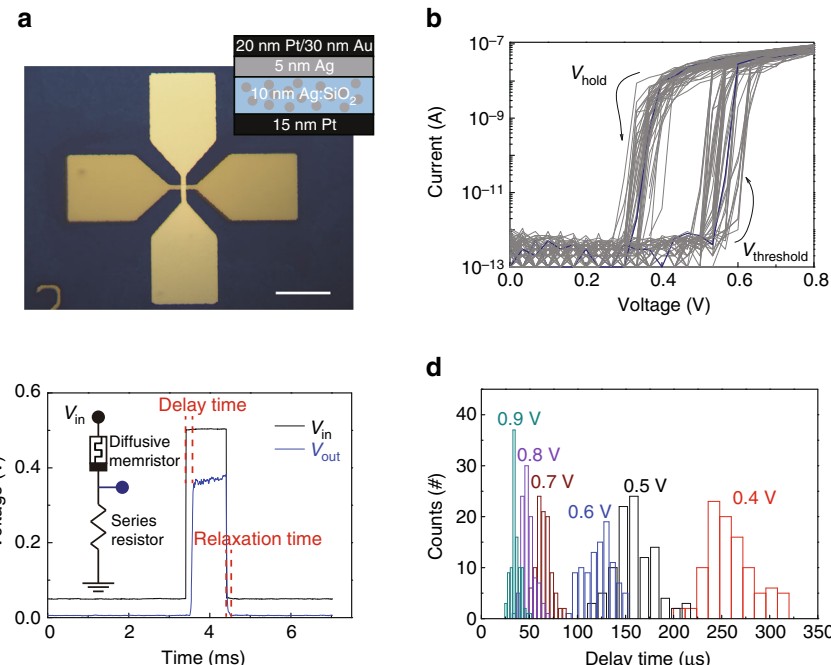

**Fig. 1** Stochastic threshold switching behavior in a Ag:SiO₂ based diffusive memristor. **a** Optical microscopic image of the 5 × 5 μm² Ag:SiO₂ cross-point device. *Scale bar*, 50 μm. *Inset* shows geometry of the Ag:SiO₂ diffusive memristor. Note that a 5 nm Ag layer is inserted between switching layer and top electrode to provide enough Ag supply. **b** 50 Consecutive DC switching cycles of the diffusive memristor connected to a 4.7 MΩ series resistor. **c** Typical pulse switching behavior of the diffusive memristor. Under a voltage pulse (300 μs in this case), some delay time is needed before the device abruptly turns ON. Inset shows the circuit for the measurements with a 120 kΩ resistor connected in series to the memristor. **d** Distribution of delay time for different input pulse amplitude (0.4 to 0.9 V at 50 Hz). A higher voltage leads to a shorter average delay time with a narrower distribution

TRNG pass all 15 NIST Special Publication 800-22 randomness tests without any post-processing. Utilizing nanoparticle dynamic simulations and simple analytical estimates, we reveal for the first time that the stochasticity in delay time originates from the stochastic process by which Ag particles detach from a Ag reservoir before their transportation to form the conduction channel(s) within $SiO_2$ matrix. The new mechanism based on ionic/atomic motion indicates that our TRNG may be less vulnerable to environmental variations such as radiation relative to other electron-based TRNGs[36].

## Results

**Stochastic volatile switching behavior of a $Ag:SiO_2$ diffusive memristor.** The optical image and geometry of a $Ag:SiO_2$ based 5 $\mu m \times 5 \mu m$ cross-point diffusive memristor used in this work is schematically shown in Fig. 1a. The device has a $Pt/Ag/Ag:SiO_2/Pt$ stack with another 30 nm thick Au on the top electrode for better contact with measurement probes (see Methods for device fabrication details). Unlike in diffusive memristors used for other applications[34], an extra Ag layer (5 nm) was inserted between the switching layer and the top electrode as a reservoir of Ag atoms to avoid any Ag depletion during switching. After fabrication, the Ag doping ratio in the $Ag:SiO_2$ switching layer was determined to be around 17% (atomic ratio) by X-ray photoelectron spectroscopy (XPS; Supplementary Fig. 1a). According to bright-field transmission electron microscopy (TEM) analysis of a 10 nm Ag:

$SiO_2$ layer deposited on a thin $SiN_x$ membrane fabricated in the same batch, dense Ag nanoclusters (mostly 2 to 5 nm in diameter with a few outliers of 10 nm) were uniformly dispersed in the $SiO_2$ matrix (Supplementary Fig. 1b).

The $Ag:SiO_2$ device did not require electroforming and exhibited reliable threshold under quasi-DC sweeps with a $> 10^5$ ON/OFF window, a sub-100 nA operation current and an extremely low (< pA) leakage current at OFF state (Fig. 1b). The device abruptly reached a low resistance state at a threshold voltage of around 0.5 V (ON-switching), followed by a spontaneous relaxation back to the high-resistance state when the voltage swept back to below 0.3 V (self-OFF-switching), confirming the volatility of the device. Multiple switching sweeps of the device also showed evident cycle-to-cycle variations in threshold voltage, verifying the stochastic nature of the switching behavior. A series resistor was found effective in limiting the ON-state current and tuning the ON/OFF window (Supplementary Fig. 2).

Stochastic delay time was observed before the sudden increase in device conductance during ON-switching under an electric pulse (Fig. 1c). A 300 $\mu s$ pulse of 0.5 V ($V_{in}$) was applied to the device, and the voltage across the series resistor ($V_{out}$) was monitored by an oscilloscope. Under this specific applied $V_{in}$, a finite delay time (incubation period, ~ 130 $\mu s$) was required before $V_{out}$ abruptly increased, indicating ON-switching of the device. When the applied voltage was removed, the device relaxed to the OFF state within 100 $\mu s$, as read by a subsequent 50 mV pulse. Figure 1d shows the statistics of the measured delay time under

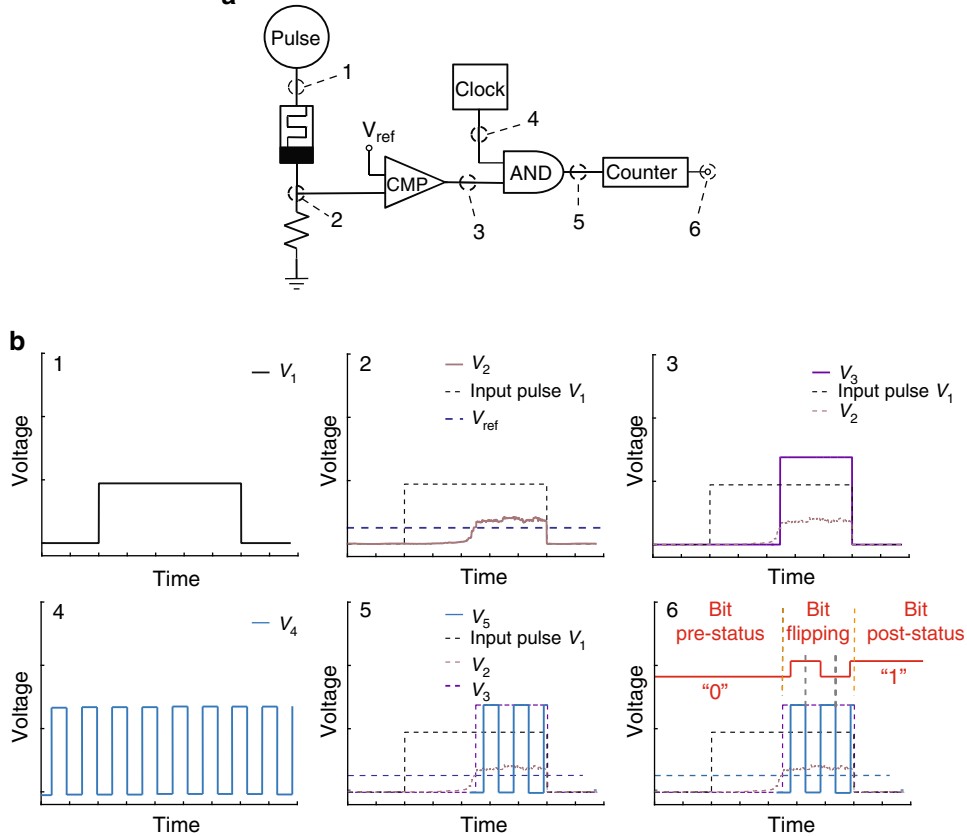

**Fig. 2** Working principle of a diffusive memristor based true random number generator (TRNG). **a** Circuit diagram of a TRNG with a diffusive memristor, a comparator (CMP), an AND gate, and a counter. **b** Schematic pulse waveforms at each stage of the circuit (as labeled in **a**), illustrating the working principle of our diffusive memristor TRNG. The stochastic delay time of the diffusive memristor leads to variations of the pulse width (shown in 3) and then random numbers of clock pulses that are sent to the counter (shown in 5). The bit status (counter output) before, during and after flipping is labeled in *red* in 6. The bit flipping is triggered by the rising edge of clock signal and hence the bit flipping frequency is half of the clock frequency. The counter output is random due to the random times of bit flipping, as determined by the random numbers of clock pulses sent to the counter ($V_5$)

different voltages (from 0.4 to 0.9 V). A higher voltage leads to a shorter average delay time with a narrower distribution. The stochastic delay time can be linked to the process of forming the Ag conduction channel(s), as will be discussed later in detail. Moreover, the delay time is also dependent on pulse frequency. As shown in Supplementary Fig. 3, a higher frequency leads to shorter delay times even with the same voltage amplitude and pulse width (0.5 V, 300 µs), which may be related to an increase of temperature of the device at higher pulse frequencies. Other factors including the speed of capacitor charging could also play a role.

**TRNG based on a diffusive memristor**. The stochastic delay time of the Ag:SiO$_2$-based diffusive memristor during ON-switching was utilized as the source of randomness for our TRNG. Figure 2a shows the circuit diagram of the proof-of-

concept unit with a diffusive memristor, a comparator, an AND-gate, and a counter. Figure 2b illustrates the operating principle of our TRNG with waveforms corresponding to each stage of the circuit as labeled in Fig. 2a. A voltage pulse ($V_1$) of fixed width is applied across a diffusive memristor and a series resistor (Panel 1). Under the applied voltage, the diffusive memristor is turned ON and hence the output voltage ($V_2$) across the series resistor suddenly increases after a stochastic delay time (Panel 2). When $V_2$ is higher than a reference voltage to the comparator ($V_{ref}$), the comparator output voltage ($V_3$) goes to a logic high level (Panel 3), and $V_2$ and $V_3$ fall back to zero when the single input pulse ($V_1$) ends. Since the delay time of the diffusive memristor is random, the comparator output voltage pulse $V_3$ has a random width. $V_3$ and a high frequency clock signal ($V_4$) are sent to an AND gate, whose output voltage pulses ($V_5$ in Panel 5) are sent to a counter. Panel 6 shows the binary bit (counter output, in red) stays at its pre-status ("0") before the device is turned ON, flips rapidly (triggered by clock signals) until the single input pulse ($V_1$) ends and then stays at its post-status ("1"). The bit flipping in the counter is triggered by the rising edge of the clock signal, and hence has a frequency half of the clock frequency. The bit on which the counter stops is random, because the stochastic delay time of the diffusive memristor leads to random pulse width from the comparator ($V_3$) and thus a random number of clock pulses that are sent to the counter. The random bit generation rate can be increased using a multi-bit counter, with which one stochastic volatile switching event can produce more than one binary bit (Supplementary Fig. 4).

The diffusive memristor TRNG was experimentally implemented by a simple circuit built on a breadboard (Fig. 3a). To demonstrate the operation of our diffusive memristor TRNG, the lowest order bit of the counter output was monitored by an oscilloscope during operation (Fig. 3b). We used a pulse train of constant amplitude ($V_1 = 0.4$ V) with pulse width of 300 and 700 µs spacing (i.e., 1 kHz frequency). As shown in Fig. 3b, the bit was initially at a low logic level ("0"). After the delay time (once the diffusive memristor switched to ON state) the counter started receiving clock signals (at 4 MHz) and the bit flipped rapidly between low and high level ("0" and "1"). At the end of the input pulse ($V_1$), the counter stopped counting and kept its last state "1" until receiving the next counting signal. This "1" was the output bit read by the microcontroller. Due to the stochastic nature of the delay time for each cycle, the counter output after each pulse was totally unpredictable and flipped randomly between "0"

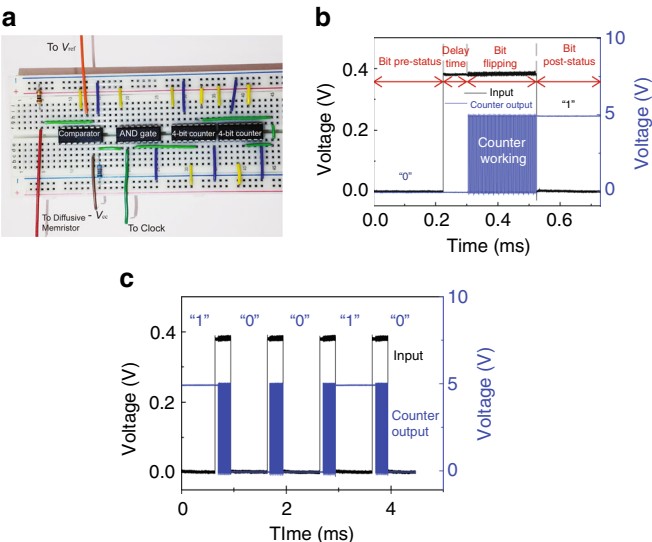

**Fig. 3** Experimental demonstration of a diffusive memristor true random number generator. **a** Photo of our simple circuit built on a breadboard. **b** Monitored one counter output in response to input voltage pulse (1 kHz) applied on our diffusive memristor. **c** Monitored one random binary output flipping from "1" → "0" → "0" → "1" → "0" over continuous switching cycles

**Table 1 Randomness test (NIST 800-22 test suite) results for a diffusive memristor true random number generator**

|  | *P*-value | Pass rate | min. pass rate | Success/failure |
|---|---|---|---|---|
| 1. Approximate entropy | 0.00983 | 75/76 | 72/76 | Success |
| 2. Block frequency | 0.768138 | 75/76 | 72/76 | Success |
| 3. Cumulative sums | 0.046525, 0.426525 | 73/76, 74/76 | 72/76 | Success |
| 4. FFT | 0.739918 | 75/76 | 72/76 | Success |
| 5. Frequency | 0.477737 | 74/76 | 72/76 | Success |
| 6. Linear complexity | 0.350485 | 76/76 | 72/76 | Success |
| 7. Longest run | 0.042413 | 76/76 | 72/76 | Success |
| 8. Non overlapping template | - | 11052/11248 | 10656/11248 | Success |
| 9. Overlapping template | 0.592591 | 75/76 | 72/76 | Success |
| 10. Random excursions | - | 360/368 | 344/368 | Success |
| 11. Random excursions variant | - | 818/828 | 774/828 | Success |
| 12. Rank | 0.094936 | 76/76 | 72/76 | Success |
| 13. Runs | 0.042413 | 75/76 | 72/76 | Success |
| 14. Serial | 0.739918, 0.795464 | 76/76, 76/76 | 72/76 | Success |
| 15. Universal | 0.000954 | 76/76 | 72/76 | Success |

Total 76M binary bits are collected from our diffusive memristor TRNG and then divided into 76 sequences (1M bits each). Tests are considered passing if *P*-value (except non-overlapping-template and random excursions variant) is > 0.0001 and the pass rate exceeds the minimum pass rate for each test. Our diffusive memristor TRNG passed all the 15 tests without any post-processing, confirming its reliable performance

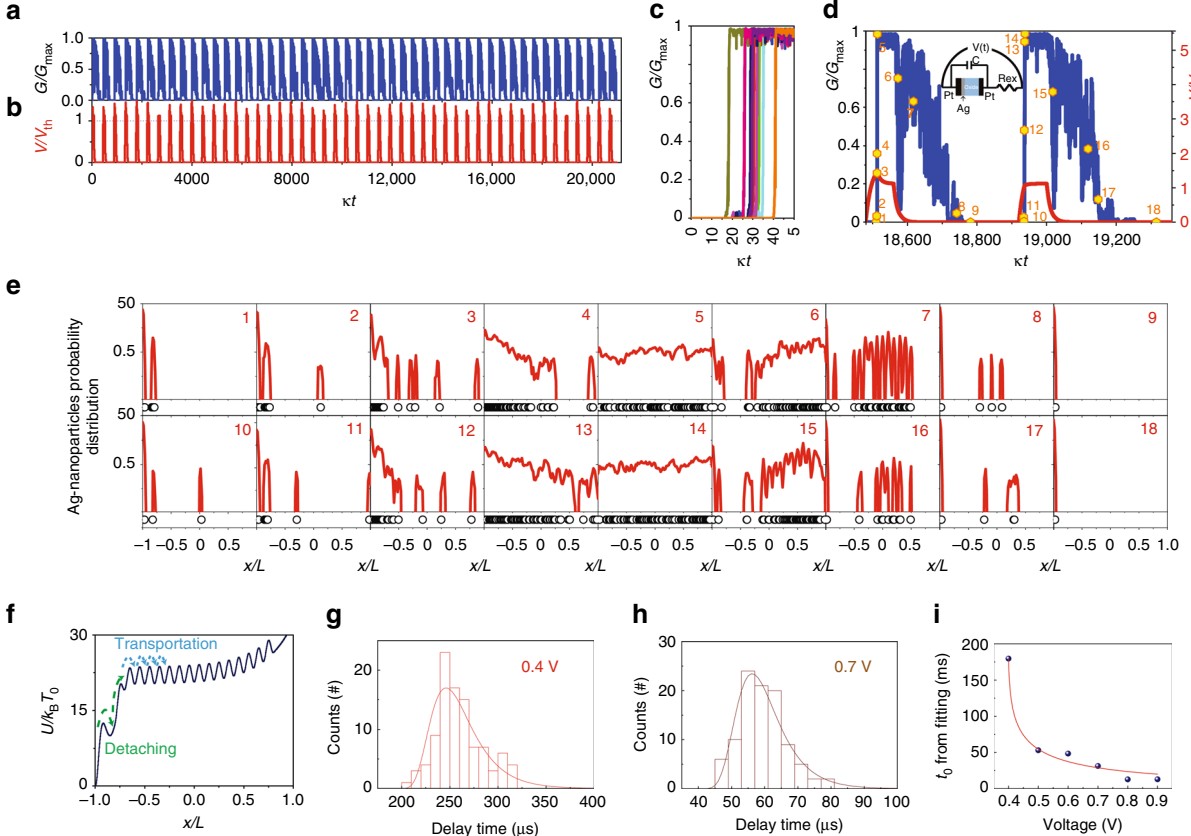

**Fig. 4** Physical origin of stochastic delay time clarified by nanoparticle dynamics simulations. **a** The switching of the simulated memristor conductance when 48 rectangular voltage pulses are applied, with conductance normalized by the maximum memristor conductance, and **b** the variation of voltage across the memristor, normalized by the threshold voltage $V_{th}$. **c** The switching to the low resistive state at time measured from the beginning of the corresponding pulses. The randomness of the resistive switching is clearly seen. **d** The two chosen resistive switches with different delay time and **e** corresponding particle probability distributions (1–18) marked on **d** by *yellow points*. *Inset* in **d** shows the circuit model used during the simulation. The memristor is connected with a parallel capacitor and a series resistor. **f** Potential normalized by thermal fluctuations across the sample used here. The delay time is composed of charging time of capacitor, time of Ag particles detach from Ag reservoir and the Ag transportation time until the formation of conduction channel(s), while the stochasticity is mainly attributed to the stochastic detaching process. For simulations in **a–e**, we used the following voltage pulse parameters: voltage pulse duration $\kappa t_p = 80$ (allowing enough time to switch to low resistive state for every pulse), inter-pulse interval $\kappa \Delta t = 360$ (allowing enough times to relax) and voltage amplitude $V_{am}/V_{th} = 1.6$, potential versus temperature as in **f** (all times measured in unit of thermal relaxation time $1/\kappa$). The experimental delay time distributions under (**g**) 0.4 V and (**h**) 0.7 V fitted by eq. 2. **i** The fitting curve of $t_0$ vs. pulse amplitudes according to eq. (3). The fitted probability distributions appear to be consistent with experimental results, confirming the feasibility of our proposed mechanism

and "1". Figure 3c shows the monitored binary bits randomly flipped from "1" → "0" → "0" → "1" → "0" during four continuous ON-switching cycles.

To assess the performance of our diffusive memristor TRNG, we carried out randomness testing for 76 million binary bits using the standard statistical testing package developed by the National Institute of Standards and Technology (NIST 800-22 test suite)[32]. We used a microcontroller's built-in 16-bit counter with 11.0592 MHz crystal oscillation frequency as the clock signal and the 6 lowest-order bits were collected. Each input pulse will give us 6 random binary bits, for a total bit generation rate of $6\,kbs^{-1}$. According to the test protocol, 76 million bits were collected (see Methods for more details) and divided into 76 sequences (1 M bits each) for the NIST tests, which returned two statistics, P-value (except non-overlapping-template and random excursions variant) and pass rate. The bits are considered random and successfully pass the test only if the P-value is greater than 0.0001 and the pass rate exceeds the minimum value defined by NIST. As shown in Table 1, our diffusive memristor TRNG passed all 15 NIST tests without any post-processing.

**Physical origin of stochastic switching delay time in Ag:SiO₂ diffusive memristors**. To better understand the physical origin of the delay time and confirm its stochastic nature, we performed nanoparticle dynamical simulations utilizing a generalized model that links electrical, nano-mechanical and thermal degrees of freedom (see Methods). Unlike previous studies that mainly focused on response to a single pulse[34, 35], multi-switching-cycles under a train of pulses were simulated. In addition to the equations for heat and Ag-nanoparticle dynamics used in previous model[34, 35], we also took into account both external and intrinsic memristor capacitances[37, 38] in order to better resemble the real conditions during experiments (see also Methods section). Figures 4a, b show the simulation results of 48 volatile switching cycles: switching of conductance G (normalized by its maximum value) is shown in Fig. 4a, while the voltage across the memristor is shown in Fig. 4b. With the same input pulses, random delay time during ON-switching was clearly observed in Fig. 4c (with time counted from the moment when the corresponding voltage pulse was applied). The delay time statistics from simulations also confirm a shorter delay time and a narrower distribution with

increasing applied voltage amplitudes (Supplementary Fig. 5a), similar to experimental results shown in Fig. 1d. Moreover, the obtained distribution statistics of simulated and measured delay times are very similar.

Evolution of Ag nanoparticle density distribution during two typical switching cycles (with $G(t)$ presented in Fig. 4d) with quite different delay times was shown in Fig. 4e (panels 1–18) step by step. A simple circuit model used during the simulation is schematically shown as an inset in Fig. 4d, which consists of a memristor, a parallel capacitor and a series resistor. The capacitor represents capacitances from the device itself and also from the external circuits during electrical measurements such as cables and breadboards. As mentioned above, a series resistor is used to limit the operation current. When a voltage is applied, the voltage across memristor gradually increases (Fig. 4d, the red curve). As soon as the voltage exceeds the threshold $V_{th}$, Ag nano-particles can randomly detach from the Ag-electrode and form a large cluster nearby (Fig. 4e, panel 1). Some particles escape from the cluster and start to diffuse towards the right electrode (Pt) driven by the electric field (Fig. 4e, panel 2) and the memristor resistance starts to decrease resulting in higher energy dissipation. This results in positive-feedback: more particles travelling towards the Pt electrode enhance the device conductance, so the heat dissipation and temperature increase activating even more particle diffusion towards the Pt electrode. Finally, some Ag particles arrive at the Pt electrode and more and more particle start to diffuse from the large left cluster (Fig. 4e, panel 3). After that, Ag nanoparticles gradually migrate towards the other terminal (forming bridge spans, Fig. 4e panel 4) and the device resistance continuously drops (and its conductance rises, Fig. 4d) until the formation of Ag conduction channel(s) that bridge the two terminals and bring the device to ON state (Fig. 4e, panel 5). After the voltage pulse is off, the capacitor gradually discharges and the voltage across the memristor decays (Fig. 4d). The conduction channel breaks (Fig. 4e, panel 6), then is further fragmented (Fig. 4e, panel 7), and Ag particles are gradually absorbed by the Ag electrodes driven by interfacial energy minimization (Fig. 4e, panels 8, 9), bringing the device to OFF state. For the case of a shorter delay (second cycle in Fig. 4d), particles detach faster and start to diffuse just after detaching from the Ag-electrode (Fig. 4e, panels 10–14), while the relaxation part of the cycle (Fig. 4e, panels 15–18) is almost the same as was described above.

Figure 4f shows the potential profile used for the simulation, which includes two energy scales: the interfacial energy responsible for detaching the Ag-electrode and formation of large metallic clusters near the device terminal and a weaker nanoparticle-pinning energy with many smaller wells between the electrodes. Based on our experiments and simulations, we have found that the delay time during ON switching consists of three steps: (i) the formation of a voltage across the device terminal (charging capacitor), (ii) Ag nanoparticles escaping from the big potential wells associated with interfacial energy (see Discussion of the detaching mechanisms in Supplementary Note 1), corresponding to detaching from the left Ag reservoir/large Ag-cluster (Fig. 4e, panel 1–2) and then (iii) transportation through the small pinning wells to the other terminal until the formation of Ag bridge(s) (Fig. 4e, Panel 2–4). The time needed to charge the capacitor (i) is deterministic in nature and delays setting voltage across the device, while the time for Ag transportation (iii) is very short (from spot 2 to spot 4, due to a fast increase in device temperature and, thus, very fast diffusion) and in practice can be neglected. Hence the stochasticity in delay time should be attributed mainly to the Ag detaching process (ii). The escape time ($t_e$) of a particle from the primary potential well (interfacial barrier) is naturally random and its distribution can be estimated

by solving the Fokker-Plank equation with parabolic well and delta-function (for classical) or ground state (for quantum) initial distribution of Ag-nanoparticles in the potential minima (Supplementary Note 1, Supplementary Figs. 6 and 7). In classical limit the delay time distribution has the form:

$$P(t_e) = \frac{Ce^{-\frac{A}{1-e^{-2kt_e}}}}{\sqrt{1-e^{-2kt_e}}} \left(e^{2kt_e}-1\right)^{-1} \tag{1}$$

Where $A$ and $k$ are fitting parameters related to potential curvature and depth of the well, and $C$ is a normalization constant. Adding the deterministic RC effect (Supplementary Note 2), one can easily get the distribution of stochastic delay time ($t$):

$$P(t) = \frac{Ce^{-\frac{A}{1-e^{-2k(t-t_0)}}}}{\sqrt{1-e^{-2k(t-t_0)}}} \left(e^{2k(t-t_0)}-1\right)^{-1} \tag{2}$$

$$t_0 = -\tau_0 \ln\left(1-\frac{V_{tr}}{V}\right) + t_1 \tag{3}$$

where $\tau_0$ is characteristic "RC" time, $V_{tr}$ is a threshold when the memristor can switch to its low resistance state if $V(t) > V_{tr}$, and $t_1$ is associated with any other deterministic voltage-independent delays (e.g., characteristic temperature relaxation time). Figure 4g, h show the fitting results for the distribution of delay time under 0.4 and 0.7 V from experiments based on eq. (2) while Fig. 4i shows the relationship between $t_0$ and applied voltages with a curve fit based on eq. (3). Similarly, we obtain a very good agreement for the simulated distribution of delay time, and RC deterministic time delays $t_0$, thus, justifying a good agreement between experimental and simulated data (Supplementary Fig. 5b). Both experimental and simulation results suggest that the stochastic process of Ag atoms detaching from Ag reservoir is responsible for the stochasticity in delay time during ON-switching. We have also checked (Supplementary Note 3 and Supplementary Fig. 8) if diffusion in higher dimensions (3D) can qualitatively change the described above simple physical picture and concluded that the mechanism described is valid until the electric field is not applied perpendicular the conducting paths.

## Discussion

Most previous approaches utilizing switching variations to build TRNGs have defined a threshold value for some switching characteristic (for example, SET voltage[30] or read current[39]). The circuit will output 1 if the measured value exceeds the threshold, and 0 otherwise. Complicated feedback and post-processing (such as von Neumann corrections) are needed to correct the ratio of 1 to 0 s (bias) and improve the randomness before running NIST tests. This is because the distribution of bits generated is highly dependent on the exact distribution of the measured character-istic. If the median value shifts over time, then 0 and 1 will not be equally probable. A better approach is desired to more efficiently exploit variations in switching characteristics as sources of ran-domness. Our method is distinct from the previous threshold-value approaches. As shown in Fig. 2, the measured delay time is used to determine how many clock pulses are sent to a counter. As a result, the mapping between analog delay time and a binary value implemented by our circuit is highly chaotic when clock frequency is fast. Each generated bit is very sensitive to even small variations in delay time, which makes the reliability of our approach immune to global shifts/drifts in the delay time dis-tribution with fast enough clock signal. Our method is a pro-mising way to exploit intrinsic stochasticity in memristive devices for security applications.

Pulse parameters were carefully chosen to optimize bitrate and ensure the randomness of generated bits. We simulated three possible cases during continuous pulse switching and identified two ways in which random number generation could fail: (i) no switching to the low resistive state during pulse, or (ii) insufficient time to relax to the high resistive state during the inter-pulse interval (Supplementary Fig. 9). Type (i) failure can be prevented by choosing a sufficiently large pulse width and amplitude, such that the device will turn ON every cycle. Type (ii) failure can be avoided by increasing the interval between pulses, so that the device is always fully relaxed and nonzero delay time occurs during every ON-switching cycle. This suggests that the pulse frequency and duration need be optimized to achieve the best performance of the diffusive memristor TRNG.

Randomness from volatile switching with a high ON/OFF ratio is easier to exploit than small noises or current fluctuations. As a result, simpler circuits are required. Our diffusive memristor TRNG can be built into memory subsystems, reducing chip area and increasing security. As revealed by simulations, the stochasticity is derived from the process of ionic motion, which suggests that our diffusive memristor TRNG (like all other memristive-switching-based TRNG) could be more resistant to harsh environments than other electron-based TRNGs[36]. Moreover, varying temperature, when TRNG is operating, affects the distribution by shifting its maximum towards lower time delays (see experiments and simulations, Supplementary Fig. 6). However, the high frequency clock used to generate random numbers makes sure the randomness is not affected.

Current bit generation rate from our diffusive memristor TRNG is $6 \, kb \, s^{-1}$, which is sufficient for many encryption applications such as internet secure session link (SSL) keys, car keys, and identification cards[9, 29]. To further increase the bit rate for other applications, a counter with more bits can be used in the circuit. Supplementary Fig. 10 shows the comparison between frequency counts of 8-bit block values if the 8-lower-order bits each cycle are collected and those of 6-bit block values from only 6-lowest-order bits. Clearly, frequency counts of 6-bit block values are uniform but those for the 8-bit block values are not, which is because those two higher-order bits (7th and 8th) do not flip frequently enough under the current clock signal. The 8th lowest bit flips 4 times slower than the 6th lowest bit. To make sure the two higher order bits also can flip more, we need to increase the clock frequency by a factor of about four ($11.0592 \times 4 = 44.2368 \, MHz$) for an 8-bit counter. In general, a higher frequency clock signal is needed so that higher order bits can also flip frequently to guarantee the generation of high quality random bit streams. This comes at the cost of increased power consumption. Alternatively, our diffusive memristor TRNG can be combined with a linear-feedback shift register (LFSR) for higher bit rates with little increase in power consumption. We performed simulation with MATLAB and demonstrated the bitrate can be readily increased by 50 times (to $300 \, kb \, s^{-1}$) with this method (Supplementary Note 4, Supplementary Fig. 11 and Supplementary Table 1). With further optimization, such as using a LFSR with more bits, the bitrate can potentially reach over 100 MHz as previously reported[29]. In addition to circuit solutions, device engineering that leads to a higher switching speed of the diffusive memristors will also improve the bitrate without changing the TRNG circuit. Possible solutions include changing the switching matrix[35, 40] or using other memristive devices. For instance, $NbO_2$ could be a good candidate for the TRNG because of its sub-nanosecond switching speed, albeit with relatively high operation current ($\sim 300 \, \mu A$) and voltage ($\sim 1.5 \, V$) even with small device size[20]. Finally, parallel operation of several diffusive memristors could also lead to increased bitrates by simultaneously generating multiple random sequences. One can efficiently save area of the circuits by 3D vertical stacking those devices[41].

We further characterize the operations of our diffusive memristor TRNG in response to two important challenges: temperature effects and long pulse cycling[39]. Our diffusive memristor TRNG still functions satisfactorily and passes the NIST tests even at 85 °C, as shown in the Supplementary Table 2. We collected 11 M binary bits under 1 kHz pulses (voltage amplitude: 0.5 V and pulse width: 300 μs). Unlike in operation at room temperature, we can only collect the 3 lowest-order bits, which means the bitrate decreases from 6 to $3 \, kb \, s^{-1}$ at 85 °C. However, this will not be a problem if we increase the clock frequency accordingly. Utilizing a clock signal of $> 8 \times 11.0592 = 88.4736 \, MHz$, one will be able to keep the bitrate steady at $6 \, kb \, s^{-1}$. The required clock frequency is dependent on the spread of delay time, and so the decreased bitrate could be a result of the decreased s.d. of delay time at high temperatures. Possible degradation due to long pulse cycling is the other concern for memristive switching based TRNG[39]. We collected 54 M binary bits from a single diffusive memristor before the device failed and got stuck at ON state (each cycle produced 6 random bits with endurance of $\sim 10^7$ cycles). Supplementary Table 3 shows that the total 54 M random bits successfully passed all 15 NIST tests. In addition, we run the tests with the first 2 M bits from the initial cycles and the last 2 M bits from the same device after $\sim 9 \times 10^6$ continuous cycles (Supplementary Table 4). Both of them passed all 15 tests, which strongly suggests that the randomness in memristors is still sufficient to generate high quality random bits even after long cycling using our method and again highlights the feasibility and novelty compared with previous attempts to build memristive switching based TRNG[29–31, 33, 39].

In conclusion, we proposed and experimentally demonstrated a novel diffusive memristor true random number generator (diffusive memristor TRNG) utilizing the stochastic delay time as the source of randomness. Our diffusive memristor TRNG has a simple structure, and shows evident advantages in circuit complexity, scalability, and power consumption. Binary bit sequences generated by our diffusive memristor TRNG passed all the 15 NIST Special Publication 800-22 randomness tests without any post-processing, a first for hardware utilizing memristive switching. Efficient approaches to further improve the bit generation rate are discussed. Finally, the physical nature of stochastic delay time during ON-switching in these devices was explained by nanoparticle dynamics simulation and simple analytical estimates and attributed to the ionic/atomic motion process, indicating our diffusive memristor TRNG could be immune to harsh environments in contrast to other electron-based TRNGs. This is the first time that the true randomness of switching variability in memristors has been confirmed with a standard test suite, which paves the way for the adoption of memristors for more security applications for the era of the IoT.

## Methods

**Device fabrication**. We used Si wafers that have 100 nm thermally grown $SiO_2$ on top as the substrates. For the $5 \times 5 \, \mu m^2$ micro-devices, the bottom electrodes were patterned by ultraviolet photolithography. After that, a 1.5 nm thick Ti adhesion layer and a 15 nm thick Pt bottom electrode were deposited sequentially in an electron beam evaporator, followed by a lift-off process in acetone. A 10 nm Ag:$SiO_2$ blanket layer was prepared by radio-frequency (RF) co-sputtering from $SiO_2$ and Ag targets (power for $SiO_2$: 270 W and Ag: 12 W). Top electrodes were defined by a second photolithography step and a 15 s $O_2$ descum, metallization of 5 nm Ag using RF sputtering (100 W) and 20 nm Pt/30 nm Au deposition using electron beam evaporator and liftoff. The extra Ag layer was used as Ag ions reservoir while the Au layer was to improve the contact between pads and probe tips.

**Electrical characterization**. The DC electrical characterizations were carried out using a Keysight B1500 semiconductor parameter analyzer in a voltage-sweep

mode. Voltage pulses were generated through a Keysight 33220A function/arbitrary waveform generator while the output waveforms were monitored by a Keysight MSO-X 3104 T mixed signal oscilloscope. During all the electrical measurements, the bottom electrodes were connected to a resistance to ground while the top electrodes were biased.

**Physical characterization**. The XPS depth profile was acquired in a Physical Electronics Instruments (PHI) quantum 2000.

**NIST Randomness tests**. A microcontroller (IAP15F2K61S2) was introduced to collect a large number of bits from our diffusive memristor TRNG. To generate those sequences, input pulses (0.5 V, pulse width: 300 μs) at a frequency of 1 kHz were continuously sent to the diffusive memristor and a series resistor. We used the microcontroller's built-in 16-bit counter (11.0592 MHz) as the clock signal and collected the 6 lower-order bits (6 kb s$^{-1}$). NIST Statistical Test Suite (Special Publication 800-22) was downloaded from the NIST websites and then run in virtual Linux system machine using the GNU Compiler Collection compiler. The test suite contains 15 randomness tests and each test targets a specific aspect of randomness. Each test returned two statistics, $P$-value (except non-overlapping-template and random excursions variant) and pass rate. The bits are considered to be random if and only if the $P$-value $\geq 0.0001$ and the pass rate exceeds the minimum pass rate for each test.

**Diffusive memristor dynamical simulations**. To simulate resistive switching in the diffusive memristor, we generalize the model used in ref. [34] where electric, heat and Ag-nanoparticle degrees of freedom were considered. In contrast to ref. [34] we have also taken into account the memristor self-capacitance, which is critically important to describe the delay time distributions. The diffusion of Ag-nanoparticles is described by the Langevin equations:

$$\eta \frac{dx_i}{dt} = -\frac{\partial U(x_i)}{\partial x_i} + \alpha \frac{V(t)}{L} + \sqrt{2\eta k_B T}\zeta \qquad (4)$$

where $x_i$ describes the location of the $i$th Ag-nanoparticle, $t$ is time, and $\eta$ is the viscosity of Ag-nanoparticles. The potential profile $U(x_i)$, where Ag nanoparticles diffuse, is formed due to interfacial interactions (see Discussion in Supplementary Note 1 and Supplementary Fig. 12) attracting small particles to the Ag-electrode and to the large cluster located near the electrode as well as large number of small minima due to pinning of Ag-nanoparticles to the device inhomogeneities and SiO$_2$ matrix structure as well as repulsion from Pt-electrode. The particular shape of potential (the potential profile used in simulations is shown in Fig. 4f) does not significantly affect the result. The only important property of the potential is the large minima associated with interfacial energy comparing with both temperature and depths of multiwell pinning potentials; the repulsive potential barrier of the Pt-electrode should be strong enough to ensure relaxation to high resistive state on a reasonable time scale when no voltage is applied. The second term in the right-hand-side of eq. (4) is related to the electric bias/tilt of the potential in the electric field $\frac{V(t)}{L}$ if Ag-nanoparticles accumulate effective charge $\alpha$ (the strength of this electric force tilt of the potential used in simulation is given in Supplementary Fig. 8a (inset)). The last term in Eq. (4) represents the unbiased δ-correlated white noise $\zeta$: $\langle \zeta(t) \rangle = 0$, $\langle \zeta(0)\zeta(t) \rangle = \delta(t)$. The noise intensity is controlled by the temperature $T$ (see Fig. 4f to estimate $k_B T_0$ with the Boltzmann constant $k_B$ and the background temperature $T_0$ with respect to the potential used in the simulations). In particular, the noise is responsible for overcoming the interfacial barrier which is suppressed by the electric force (in the simulations the electric field decreases the interfacial potential well by a factor of 1.6 at $V_{th}$); this results in switching the system to its low resistance state when a voltage pulse is applied. In addition, the noise generates the diffusion of Ag-nanoparticles towards the Ag-electrode when voltage is off resulting in the thermal relaxation of the memristors.

The heat dynamics in the memristor are described by Newton's cooling law:

$$\frac{dT}{dt} = \mathbb{C}_T^{-1} Q - \kappa(T - T_0) \qquad (5)$$

where $\mathbb{C}_T$ is the memristor heat capacitance, $Q = V(t)^2/R(x_1, x_2, .., x_N)$ is Joule heat power with memristor resistance $R(x_1, x_2, .., x_N)$, which depends on Ag-nanoparticle locations, $\kappa$ is the heat transfer coefficient describing heat flux from the device. Note that the actual system temperature and the macroscopic-cluster temperature can be significantly different. We assume the resistance has a tunneling nature and is described by the equation $R(x) = R(x_1, x_2, .., x_N) = R_t \sum_0^N e^{(x_{i+1} - x_i)/\lambda}$ where $x_0$ and $x_{N+1}$ are positions of the device terminals, $R_t$ resistance amplitude and $\lambda$ is the tunneling length. The minimum possible resistance occurs when all Ag-nanoparticles are equally separated and has the value $R_{min} = (N+1)R_t e^{(x_{N+1} - x_0)/((N+1)\lambda)}$ (we used $\lambda/L = 0.12$ for 1D and 0.2 for 3D simulations).

As a distributed system with high resistance the memristor can have a capacitance ($C_M$) which was not considered in previous models[34, 35]. In general, this capacitance could be different in the two memristor states[37, 38], but we, for simplicity, assume that $C_M$ is a constant and is not a function Ag nano-particle

locations. A simple consideration of a circuit consisting of the memristor resistance connected in parallel to the memristor capacitance result in the equation for voltage drop $V(t)$ across the memristor:

$$\tau_0 \frac{dV}{dt} = V_{ex}(t) - \left(1 + \frac{R_{ex}}{R(x)}\right)V \qquad (6)$$

where "RC" time $\tau_0 = C_M R_{ex}$ with the resistance $R_{ex}$ of external wires connected in sequence with the memristor (for simulations we used $\kappa\tau_0 = 16$ and $\frac{R_{ex}}{R_{min}} = 0.5$), and $V_{ex}$ describes the applied voltage pulses.

**Data availability**. The data that support the findings of this study are available from the corresponding author upon request.

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

## Acknowledgements

This work was supported in part by the U.S. Air Force Office for Scientific Research (AFOSR) (Grant No. FA9550-12-1-0038), the U.S. Air Force Research Laboratory (AFRL) (Grant No. FA8750-15-2-0044), and the National Science Foundation (NSF; ECCS-1253073). D.B. is supported by a Research Experience for Undergraduates (REU) supplement grant from NSF. Any opinions, findings and conclusions or recommendations expressed in this material are those of the authors and do not necessarily reflect the views of AFRL. We thank C. Yu and Dr. Xiaolin Xu for helpful discussions.

## Author contributions

Q.X. and H.J. conceived the idea, Q.X., J.J.Y. and H.J. designed the experiments. H.J. performed device fabrication and engineering. D.B. and S.L. built the circuit. H.J., D.B. and S.L. performed electrical measurements. S.E.S. performed the simulation and modeling. Z.W., Y.L., S.J., R.M. and C.L. helped with experiments and data analysis. Q.X., H.J., S.E.S. and J.J.Y. wrote the manuscript. All authors discussed the results, commented on and gave approval to the final version of the manuscript.

## Additional information

**Competing interests:** The authors declare no competing financial interests.

