## [transparent peer review · Nature Communications]

Reviewer #1 (Remarks to the Author):

This manuscript reported a new random number generator based on the delay time of the switching behavior in SiO₂:Ag memristors. This work is interesting and timely to me. But, I have some comments needed to be addressed before a decision on the manuscript is made. Detailed comments are:

1 The delay time of the threshold switching is dependent on the amplitude of the applied voltage. In Figure 1d, the authors have measured the delay time under different voltage pulses. Under 0.4V pulse amplitude, the delay time distribution is from about 200us to 320us, and the mean value is about 250us. But in the following measurements (Figure 3b and 3c), the delay time is about 100us at (0.4V, 300us) voltage pulses. What is the reason? If the mean delay time is 250us, the pulse width (300us) in the following measurements is not suitable.

2 In the measurement system as illustrated in Figure 2, an 8-bit counter combined by two 4-bit counters are used, the clock frequency is 4MHz and the pulse width and interval is 300us and 700us (i.e. 1kHz frequency) (page 10, line 202~206). So, the bit generation rate is calculated to be 8kb/s. But the author claims it to be 6kb/s in the Discussion (page 18, line 368). Furthermore, the author claims in the Discussion (page 18, line 373) that a higher clock frequency (40MHz) is needed for an 8-bit counter for high quality random bit streams. These descriptions are contradiction and the author should clarify this.

3 Generally, the electrical characteristics (i.e. delay time) of the next threshold switching is likely similar to that of the current one in most cases. However, it may be more different from the switching hundreds after it. Therefore, the switching behavior of this device maybe not real randomness, please comment it.

Reviewer #2 (Remarks to the Author):

The paper reports research results about true random number generator (TRNG) by the stochastic delay of switching in a Ag-based memristor. Although the paper is well presented and clearly written, but the technical content remains insufficient for publication with current version. Several questions are listed below for authors,

1. Authors mentions that one of the advantages of the memristor based TNG is applied in harsh environments, I am wondering that what is the temperature dependance of this TNG, will the frequency of the data generation rate will be affected by high or low temperatures? Will the high/low randomness data still pass NIST's test in the long run and frequency items?

2. Fig.1 and Supplementary Figure 2 show the switching variation is quite wide, after a serial of on/off switch (cycling) or long term storage at On or Off state the randomness level could be close or mixed, do authors have characterization show the robustness of the memristor?

3. According to the discussion of TRNG based on a diffusive memristor in page 8 and the data showing in Fig.3, the bit pre-status and delay time is quite long, moreover the bit flipping over takes 2ms to finish it, what is the bottleneck of the time-consuming process? Will it affect the output rate of randomness data in NIST test (Frequency)?

4. Regarding the discussion of Physical origin of stochastic switching delay time in Ag:SiO₂ diffusive memristor in Page 13, the authors mentioned that the delay time is composed of charging time of capacitor, time of Ag particles detach from Ag reservoir and the Ag transportation time until the formation of conduction channel, while the stochasticity is mainly attributed to the stochastic detaching process. Including the potential well derivation in

Supplementary Note 1, the question is the possibility of Ag detaching from well is strongly correlated with the height of potential well, how to estimate the energy barrier? And do the authors include 3D quantum dot model in this estimation to make it more realistic? The detaching probability should not be significantly affected by thermal energy, is that matching with experimental result (compare to the reviewer's question 1)?

Reviewer #3 (Remarks to the Author):

The work presents a new solution for random number generation with a diffusive memristor device, consisting of a nanoionic switching element with unstable low-resistance state. The solution for number generation is new, however it shows some limitations in performance. In particular, the solution does not overcome one of the limitation that is mentioned in the introductory part of the manuscript, namely that 'need for complicated probability tracking and careful tuning of the applied voltage/current'. In fact, to reach the random delay time in a desired time range, one should carefully tune the applied voltage to achieve that mean value of delay time. Also, the frequency operation of such TRNG is limited by the relaxation time (e.g. 0.1 ms in Fig. 1c), since it is necessary to wait for the relaxation of the Ag particles after every switching process. The fact that the technique passed MNIST test is important, although other constraints (bandwidth limitation, probability tracking, which are critical for this solution) are not considered in the MNIST metric. The claim that this solution is inherently radiation hard is valid for all memristor-based number generators, not only the present one. Numerical simulations are added to support the mechanism, but the model and explanation are basically the same as in previous works by the same authors [34, 35]. Overall I perceive the advance in this work as quite narrow and incremental, which might make it better suited for a specialized journal of circuits and systems, after the probability tracking and bandwidth limitation are carefully taken into account.

We would like to thank the reviewers very much for the constructive comments on our manuscript. Per requests from the reviewers, **we have conducted additional electrical measurements on frequency and temperature dependency, run extra NIST tests, and performed new numerical simulations. We have also discussed our technology as compared with others and clarified some possible confusions in the text.** We revised both the main text and the supplementary information (SI) accordingly based on the newly acquired data. In the following point-to-point response to the reviewers' technical comments, the original comments are in black fonts, our responses are in blue fonts. Changes in the revised manuscript main text and SI are also highlighted in blue fonts.

Response to Reviewer #1:

This manuscript reported a new random number generator based on the delay time of the switching behavior in SiO₂:Ag memristors. This work is interesting and timely to me. But, I have some comments needed to be addressed before a decision on the manuscript is made.

RESPONSE:

We thank the reviewer for the positive comment on the novelty and timing of our work. We will address the technical comments in the next few paragraphs.

Detailed comments are:

1 The delay time of the threshold switching is dependent on the amplitude of the applied voltage. In Figure 1d, the authors have measured the delay time under different voltage pulses. Under 0.4V pulse amplitude, the delay time distribution is from about 200us to 320us, and the mean value is about 250us. But in the following measurements (Figure 3b and 3c), the delay time is about 100us at (0.4V, 300us) voltage pulses. What is the reason? If the mean delay time is 250us, the pulse width (300us) in the following measurements is not suitable.

RESPONSE:

This is an important point, we thank the reviewer for pointing this out. The delay time of the diffusive memristor is also dependent on the pulse frequency. Figure 1d is a typical distribution of delay time as a function of pulse amplitude at 50 Hz, while results in Fig. 3 are from a 1 kHz pulse train that was used to demo the TRNG principle. We found out that under the otherwise identical conditions (e.g., pulse width, amplitude), a higher frequency leads to a shorter delay time. We have added this data to the revised Supplementary Information (as new Supplementary Fig. 3). A possible explanation of this effect is that average current passing through the memristor increases, leading to an increased temperature in the device. A higher temperature results in a shift of the distribution to smaller delay time as seen in the presented figure.

Moreover, other factors such as the speed of capacitor charging may also play a role depending on the frequency range and materials system. To clear possible confusion, frequencies used for Fig. 1d and Fig. 3 are also specified in the figure captions in the revised manuscript. In addition, we added the following sentence in the main text (page 8, paragraph 1) that states the frequency dependency.

“Moreover, the delay time is also dependent on pulse frequencies. As shown in Supplementary

Fig. 3, a higher frequency leads to shorter delay times even with the same voltage amplitude and pulse width (0.5 V, 300 μ s), which may be related to an increase of temperature of the device at higher pulse frequencies. Other factors including the speed of capacitor charging could also play a role.”

Supplementary Figure 3 | Frequency dependency of delay time. (a) Distributions of delay time at different frequencies (from 50 Hz to 1kHz) for electrical pulses with the same voltage amplitude and pulse width (0.5 V, 300 μ s). (b) Plot of median delay time vs. frequency. A higher frequency leads to a shorter median delay time.

2 In the measurement system as illustrated in Figure 2, an 8-bit counter combined by two 4-bit counters are used, the clock frequency is 4MHz and the pulse width and interval is 300us and 700us (i.e. 1kHz frequency) (page 10, line 202~206). So, the bit generation rate is calculated to be 8kb/s. But the author claims it to be 6kb/s in the Discussion (page 18, line 368). Furthermore, the author claims in the Discussion (page 18, line 373) that a higher clock frequency (40MHz) is needed for an 8-bit counter for high quality random bit streams. These descriptions are contradiction and the author should clarify this.

RESPONSE:

Thank the reviewer for the suggestions. For experimental demonstration of the working principle of our diffusive memristor TRNG in Fig. 2, although we used an 8-bit counter (created by combining two 4-bit counters), we only monitored the single lowest-order bit using an oscilloscope to showcase how it randomly flips and experimentally demonstrate our proposed circuit. For later large number of binary bits, it is more convenient to use the microcontroller’s built-in 16-bit counter with 11.0592 MHz crystal oscillation frequency. And we collected the 6 lowest-order bits. Consequently, each input pulse will give 6 random binary bits and hence the total bit generation rate is 6 kb/s instead of 8 kb/s (as explained in Methods/ NIST Randomness Tests). The reason why only the 6 lower-order bits were collected is that the 2 higher-order bits may not flip frequently under the current clock signal. This is verified by our experimental results, as shown in the following figure (new Supplementary Fig. 8. Clearly, frequency counts of 6-bit block values are uniform but those for the 8-bit block values are not. To collect 8 random bits from the counter, a higher frequency clock signal is needed so that higher order bits can also

flip quickly enough. The 8th lowest bit flips 4 times slower than the 6th lowest bit. As a result, we need to increase the clock frequency at least by a factor of 4 ($11.0592 \times 4 = 44.2368$ MHz) for an 8-bit counter.

Supplementary Figure 8 | Bit frequency uniformity at different data collection scheme. (a) If each time 8-bit data is collected from the 8 lower-order bits of the counter, the block values are not uniform. (b) If only the 6 lower-order bits are collected, each block is equally likely. The data collecting circuit uses a built-in clock in the micro-controller that has a 11.0592 MHz crystal oscillation frequency. The 8th lowest bit flips 4 times slower than the 6th lowest bit and hence the experiment suggests that a ≥ 4 times higher clock signal is required for an 8-bit counter.

To avoid the misunderstanding, we deleted “*Two 4-bit counters were combined into an 8-bit counter to increase the bit generation rate.*” (page 10, paragraph 1) We are only monitoring the first lower order bit for the results in Fig. 3. It will not affect understanding of information we try to deliver in Fig. 3. We also changed “*one of the counter outputs*” (page 10, paragraph 1) to “*the lowest order bit of the counter output*” to make it clear.

To emphasize why the bitrate is 6 k/s, we moved a couple of sentences from the Methods to the main text (page 12, paragraph 1):

“We used the microcontroller’s built-in 16-bit counter with 11.0592 MHz crystal oscillation frequency as the clock signal and the 6 lowest-order bits were collected. Each input pulse will give us 6 random binary bits and the total bit generation rate is 6 kb/s.”

To explain more clearly why we choose only 6-lowest-order bits, we have also added the following sentences in the main text (page 20, paragraph 3):

“Supplementary Fig. 8 shows the comparison between frequency counts of 8-bit block values if the 8-lower-order bits each cycle are collected and those of 6-bit block values from only 6-lowest-order bits. Clearly, frequency counts of 6-bit block values are uniform but that for the 8-bit block values are not, which is because those two higher-order bits (7th and 8th) do not flip frequently under the current clock signal. The 8th lowest bit flips 4 times slower than the 6th

lowest bit. To make sure the two higher order bits also can flip more, we need to increase the clock frequency by a factor of about four ($11.0592 \times 4 = 44.2368 \text{ MHz}$) for an 8-bit counter.”

3 Generally, the electrical characteristics (i.e. delay time) of the next threshold switching is likely similar to that of the current one in most cases. However, it may be more different from the switching hundreds after it. Therefore, the switching behavior of this device maybe not real randomness, please comment it.

RESPONSE:

Thank the reviewer for bringing up this interesting point, which exactly highlights one of the advantages of our approach. We did observe that the delay time of the diffusive memristor could vary less between consecutive switching events (local) than that after hundreds of switching events (global). For example, Fig.R1a below shows the delay times from 1000 consecutive switching events. The variation of delay time within either zone 1 (gray) or zone 2 (blue) is much smaller than the average variation between these two zones. However, the global shift in the delay time is unpredictable (i.e., random). A detailed analysis for zone 2 shows that the local distribution of the delay time is also random (Fig. R1b). As demonstrated, our method is not affected by the global shift in the delay time since even small variations can lead to stochastic changes in the number of clock pulses sent to the counter and hence random output, as long as the scale of these variations is comparable to the clock frequency. This further confirms that our method is a promising way to exploit intrinsic stochasticity in memristive devices for security applications.

Figure R1 | Global shift in delay time. (a) Delay times from 1000 consecutive cycles (pulse amplitude: 0.5 V, pulse width: 300 μs and frequency: 1 kHz) and (b) magnified view of zone 2. The median value of the delay time is 71 μs, as labelled by the blue line in (a).

The following paragraph has been added into the main text (page 19, paragraph 2):

“Most previous approaches utilizing switching variations to build TRNGs have defined a threshold value for some switching characteristic (for example SET voltage³⁰ or read current⁴¹). The circuit will output 1 if the measured value exceeds the threshold, and 0 otherwise. Complicated feedback and post-processing (such as von Neumann corrections) are needed to correct the ratio of 1s to 0s (bias) and improve the randomness before running NIST tests. This is because the distribution of bits generated is highly dependent on the exact distribution of the measured characteristic. If the median value shifts over time, then 0 and 1 will not be equally probable. A better approach is desired to more efficiently exploit variations in switching characteristics as sources of randomness. Our method is distinct from the previous threshold-value approaches. As shown in Fig. 2, the measured delay time is used to determine how many clock pulses are sent to a counter. As a result, the mapping between analog delay time and a binary value implemented by our circuit is highly chaotic when clock frequency is fast. Each generated bit is very sensitive to even small variations in delay time, which makes the reliability of our approach immune to global shifts/drifts in the delay time distribution with fast enough clock signal. Our method is a promising way to exploit intrinsic stochasticity in memristive devices for security applications.”

Response to Reviewer #2:

The paper reports research results about true random number generator (TRNG) by the stochastic delay of switching in a Ag-based memristor. Although the paper is well presented and clearly written, but the technical content remains insufficient for publication with current version.

RESPONSE:

We thank the reviewer for the positive comment on the presentation of our work. We will address the technical comments in the next few paragraphs.

Several questions are listed below for authors,

1. Authors mentions that one of the advantages of the memristor based TNG is applied in harsh environments, I am wondering that what is the temperature dependence of this TNG, will the frequency of the data generation rate will be affected by high or low temperatures? Will the high/low randomness data still pass NIST's test in the long run and frequency items?

RESPONSE:

We appreciate the reviewer for bringing up the temperature factor. To study the temperature effect, we collected 11 M binary bits under 1 kHz pulses (voltage amplitude: 0.5 V and pulse width: 300 μ s.) at 85 °C. As shown in the Supplementary Table 1, these bits successfully passed all the 15 NIST tests, including in the long run and frequency items (more details to follow). In contrast to operation at room temperature, we have to collect only the 3 lower-order bits, which means the bitrate decreases from 6 k/s to 3 k/s at 85 °C. However, this will not be a problem if a very fast clock signal is available. Utilizing a clock signal of $> 8 \times 11.0592 = 88.4736$ MHz, one will be able to keep the bitrate staying at 6 k/s. The required clock frequency is dependent on the spread of delay time and hence the decreased bitrate could be possibly attributed to the smaller standard deviation of delay time at high temperatures. We have also performed simulations and

measurements at different temperatures, and observed very good agreements. This also further justifies the proposed thermal activation mechanism of the random memristor switching when particle hopped over the interfacial barrier detaching from the Ag-electrode.

	P -Value	Pass Rate	min. pass rate	SUCCESS / FAILURE
1.Approximate Entropy	0.275709	11/11	9/11	SUCCESS
2.Block Frequency	0.090936	11/11	9/11	SUCCESS
3.Cumulative Sums	0.834308 , 0.637119	11/11, 11/11	9/11	SUCCESS
4.FFT	0.025193	11/11	9/11	SUCCESS
5.Frequency	0.162606	11/11	9/11	SUCCESS
6.Linear Complexity	0.437274	10/11	9/11	SUCCESS
7.Longest Run	0.437274	11/11	9/11	SUCCESS
8.Non Overlapping Template	-	1608/1628	1332/1628	SUCCESS
9.Overlapping Template	0.834308	11/11	9/11	SUCCESS
10.Random Excursions	-	72/72	64/72	SUCCESS
11.Random Excursions Variant	-	156/162	144/162	SUCCESS
12.Rank	0.637119	11/11	9/11	SUCCESS
13.Runs	0.090936	10/11	9/11	SUCCESS
14.Serial	0.637119 , 0.437274	11/11, 11/11	9/11	SUCCESS
15.Universal	0.637119	10/11	9/11	SUCCESS

Supplementary Table 1 | Randomness test (NIST 800-22 test suite) results for a diffusive memristor true random number generator working at 85 °C. Total 11 M binary bits were collected and passed all the 15 tests with no post processing. Compared to operations at room temperature, the bitrate decreased to 3 kb/s since we can only collect the 3 lower-order bits from the counter with the same 11.0592 MHz crystal oscillation frequency. The bitrate can be kept at 6 kb/s if ~ 88.4736 MHz clock signal is used.

Since the reviewer is interested in the Longest Run and Frequency items and these two tests don't need as much as 1 M bits, we divided 11 M bits into 110 sequences (100 k each) and ran these two tests again. The results are shown in Table R1.

	P -Value	Pass Rate	min. pass rate	SUCCESS / FAILURE
Frequency	0.684327	109/110	105/110	SUCCESS
Longest Run	0.757969	110/110	105/110	SUCCESS

Table R1 | Frequency and Longest Run results for a diffusive memristor true random number generator working at 85 °C. Total 11 M binary bits were divided into 110 sequences (100 k each).

2. Fig.1 and Supplementary Figure 2 show the switching variation is quite wide, after a serial of on/off switch (cycling) or long term storage at On or Off state the randomness level could be close or mixed, do authors have characterization show the robustness of the memristor?

RESPONSE:

We thank the reviewer for this important question. The diffusive memristors still function properly even after being in storage for over half a year. From single diffusive memristor, currently we can at most collect 54 M binary bits before the device failed and got stuck at ON state (each cycle will output 6 random bits with endurance of $\sim 10^7$ cycles). Supplementary Table 2 shows total 54 M random bits successfully passed all the 15 NIST tests. In addition, we run the tests with the first 2 M bits from the very beginning cycles and the last 2 M bits from the same device that after $\sim 9 \times 10^6$ cycles (Supplementary Table 3). Both of them passed all the 15 tests, which strongly supports that the randomness in memristors is still sufficient to generate high quality random bits even after long cycling using our method. We expect this can be significantly improved further in the future after these newly developed diffusive memristors are optimized for this purpose. Even if the variance in the delay time becomes smaller after many cycles, our clock frequency is fast enough that the output is observed to be random.

	P -Value	Pass Rate	min. pass rate	SUCCESS / FAILURE
1.Approximate Entropy	0.383827	54/54	51/54	SUCCESS
2.Block Frequency	0.534146	54/54	51/54	SUCCESS
3.Cumulative Sums	0.699313 , 0.616305	53/54, 53/54	51/54	SUCCESS
4.FFT	0.657933	54/54	51/54	SUCCESS
5.Frequency	0.657933	53/54	51/54	SUCCESS
6.Linear Complexity	0.171867	53/54	51/54	SUCCESS
7.Longest Run	0.534146	54/54	51/54	SUCCESS
8.Non Overlapping Template	-	7921/7992	7548/7992	SUCCESS
9.Overlapping Template	0.236810	54/54	51/54	SUCCESS
10.Random Excursions	-	296/296	272/296	SUCCESS
11.Random Excursions Variant	-	661/666	612/666	SUCCESS
12.Rank	0.494392	11/11	51/54	SUCCESS
13.Runs	0.090936	54/54	51/54	SUCCESS
14.Serial	0.574903 , 0.108791	53/54, 53/54	51/54	SUCCESS
15.Universal	0.213309	54/54	51/54	SUCCESS

Supplementary Table 2 | Randomness test (NIST 800-22 test suite) results for 54 M binary bits from a single diffusive memristor with a bitrate of 6 kb/s at room temperature.

First 2 M Bits			Last 2 M Bits		
	P -Value	SUCCESS / FAILURE	Last 2 M	P -Value	SUCCESS / FAILURE
1.Approximate Entropy	0.285	SUCCESS	1.Approximate Entropy	0.924	SUCCESS
2.Block Frequency	0.94	SUCCESS	2.Block Frequency	0.788	SUCCESS
3.Cumulative Sums	0.388, 0.187	SUCCESS	3.Cumulative Sums	0.275, 0.235	SUCCESS
4.FFT	0.881	SUCCESS	4.FFT	0.058	SUCCESS
5.Frequency	0.201	SUCCESS	5.Frequency	0.935	SUCCESS
6.Linear Complexity	0.314	SUCCESS	6.Linear Complexity	0.195	SUCCESS
7.Longest Run	0.297	SUCCESS	7.Longest Run	0.078	SUCCESS
8.Non Overlapping Template	-	SUCCESS	8.Non Overlapping Template	-	SUCCESS
9.Overlapping Template	0.638	SUCCESS	9.Overlapping Template	0.106	SUCCESS
10.Random Excursions	-	SUCCESS	10.Random Excursions	-	SUCCESS
11.Random Excursions Variant	-	SUCCESS	11.Random Excursions Variant	-	SUCCESS
12.Rank	0.367	SUCCESS	12.Rank	0.309	SUCCESS
13.Runs	0.413	SUCCESS	13.Runs	0.585	SUCCESS
14.Serial	0.202, 0.081	SUCCESS	14.Serial	0.692, 0.763	SUCCESS
15.Universal	0.452	SUCCESS	15.Universal	0.352	SUCCESS

Supplementary Table 3 | Randomness test (NIST 800-22 test suite) results for first 2 M first 2 M bits (of the 54 M bits from the same diffusive memristor) from the very beginning cycles and the last 2 M bits from the same device that after $\sim 9 \times 10^6$ cycles. Both of them passed the tests, indicating that even after many cycles the randomness in memristive switching remains sufficient to produce high quality random bits using our method.

We thank the reviewer again for pointing out the two challenges (temperature and cycling) to memristive switching based TRNGs. We have added the following paragraph into the main text (page 22, paragraph 2):

“We further characterize the operations of our diffusive memristor TRNG in response to two important challenges: temperature effects and long pulse cycling⁴¹. Our diffusive memristor TRNG still functions satisfactorily and passes the NIST test even at 85 °C, as shown in the Supplementary Table 1. We collected 11 M binary bits under 1 kHz pulses (voltage amplitude: 0.5 V and pulse width: 300 μs.). Unlike in operation at room temperature, we can only collect the 3 lowest-order bits, which means the bitrate decreases from 6 to 3 k/s. at 85 °C. However, this will not be a problem if we increase the clock frequency accordingly. Utilizing a clock signal of $> 8 \times 11.0592 = 88.4736$ MHz, one will be able to keep the bitrate steady at 6 k/s. The required clock frequency is dependent on the spread of delay time, and so the decreased bitrate could be a result of the decreased standard deviation of delay time at high temperatures. Possible degradation due to long pulse cycling is the other concern for memristive switching based TRNG⁴¹. We collected 54 M binary bits from a single diffusive memristor before the device failed and got stuck at ON state (each cycle will output 6 random bits with endurance of $\sim 10^7$ cycles). Supplementary Table 2 shows that the total 54 M random bits successfully passed all 15 NIST tests. In addition, we run the tests with the first 2 M bits from the initial cycles and the last 2 M bits from the same device after $\sim 9 \times 10^6$ continuous cycles (Supplementary Table 3). Both of them passed all 15 tests, which strongly suggests that the randomness in memristors is still sufficient to generate high quality random bits even after long cycling using our method and again highlights the feasibility and novelty compared with previous attempts to build memristive switching based TRNG^{29-31, 33, 41}”

3. According to the discussion of TRNG based on a diffusive memristor in page 8 and the data showing in Fig.3, the bit pre-status and delay time is quite long, moreover the bit flipping over takes 2ms to finish it, what is the bottleneck of the time-consuming process? Will it affect the output rate of randomness data in NIST test (Frequency)?

RESPONSE:

This is an important point that eventually determines the application of our TRNG. A typical delay time of our device under 0.5 V (1 kHz) is tens of μs. During operation, we used 300 μs pulse width to ensure the device can be turned ON every cycle. Because of these, usually 0.2 ms (not 2 ms) is left for the bit flipping. One can increase the voltage to shorten the delay time and use narrower pulses to minimize the bit-flipping-time, but the bottleneck is the relaxation time, which is typically around 100 μs and limits the output rate. The relaxation is a process in which nanoparticles start to coalesce and eventually merge into larger clusters to minimize interfacial energy when the voltage pulse is removed. This process brings the device back to high resistance state and takes time due to its diffusive nature¹.

For the demo of our proposed diffusive memristor TRNG, we are not using an integrated circuit, which makes the switching speed we can deal with relative slow due to large parasitic effects (capacitances, for example). It is easier and also quicker for us to deal with a slow switching speed with the set-up on bread-boards as shown in the paper. Regarding the switching speed, device engineering can be used and have been successfully demonstrated to significantly short the relaxation time to hundreds of ns.^{2,3} In addition, our proposed method to build a TRNG can be easily applied to other threshold-switching devices. For instance, NbO₂ could be a good candidate because of its fast switching speed (delay time: 0.7 ns and relaxation time: 2.3 ns)⁴.

With this speed, the potential bitrate can reach more than 300 MHz even using 1-bit counter. For faster devices, a much faster clock signal is also required, which is not going to be a problem (a stable clock rate of 3 GHz has been used for the CPU (Intel Pentium 4 model) 15 years ago). Therefore, the output rate can be much faster if we adopt a fully integrated circuit.

Moreover, even our current diffusive memristor TRNG already shows promises for applications in low-power and low-speed encryption systems, such as internet secure session link (SSL) keys, car keys and identification cards^{5,6}, as discussed on page 19 in the main text. For high-frequency applications, one of traditional technique is to combine such low-frequency TRNG with a linear-feedback shift register^{5,7}, which can greatly increase the bitrate as we have demonstrated in Supplementary Note 3.

4. Regarding the discussion of Physical origin of stochastic switching delay time in Ag:SiO₂ diffusive memristor in Page 13, the authors mentioned that the delay time is composed of charging time of capacitor, time of Ag particles detach from Ag reservoir and the Ag transportation time until the formation of conduction channel, while the stochasticity is mainly attributed to the stochastic detaching process. Including the potential well derivation in Supplementary Note 1, the question is the possibility of Ag detaching from well is strongly correlated with the height of potential well, how to estimate the energy barrier? And do the authors include 3D quantum dot model in this estimation to make it more realistic? The detaching probability should not be significantly affected by thermal energy, is that matching with experimental result (compare to the reviewer's question 1)?

RESPONSE:

We conducted additional experiments and performed simulations at different temperatures (Supplementary Fig. 7). The simulations are qualitatively consistent with the experimental results and the distributions of delay time can still be fitted by the equation 2 as shown in Supplementary Fig. 7a and 7b. The measured delay time is decreasing as the temperature increases with an activation energy of ~ 0.2 eV (Supplementary Fig. 7c), in a good agreement with simulations (Supplementary Fig. 7d).

Concerning an estimation of the interfacial barrier, we developed a simple variational approach allowing to minimize Ag-nanoparticle shape in the process of detaching from the Ag-electrode (Supplementary Note 1). Even in the classical limit, we show that the barrier depends on the quality of the interface between contacting surfaces separating the electrode and the Ag-nanoparticle. For the case of a dirty interface the estimate (0.1 eV) is close to the experimental data and simulations. However, it requires an assumption about the renormalization of interfacial energy of the two contacting (attaching) Ag-surfaces. It should be noted that to fully understand the barrier would require detailed experiments that are beyond the scope of this paper.

Supplementary Figure 7 | Temperature dependence of delay time from experiments and simulations. (a) Distributions of delay time (t_{delay}) at 25 °C and 55 °C (voltage amplitude: 0.5 V and pulse width: 300 μs @ 1 kHz) can still be fitted by equation (2) in the main text, in perfect agreement with (b) simulated results (all parameters are the same as in simulations presented in the main text and temperature ratio for blue and green histograms are 5/6). (c) The delay time is decreasing as the temperature increased with an activation energy of 0.2 eV, also consistent with that in simulations in (d).

Following the referee’s suggestions, we extend our estimation to incorporate quantum effects in our analysis in Supplement Note 1. We decided still to consider 1D situation since these estimates are qualitative in any case and 3D estimations require precise knowledge of the 3D potential which is hard to be reconstructed. 1D simulations should work well if the potential has a preferable direction for particle to escape which is quite likely. The Supplementary Note 1 was hence re-written with supporting Supplementary Figs. 7 and 10. Since we observed a shift of the probability distribution maximum to the left (in Supplementary Fig. 7), which contradicts to the simulated results in Supplementary Fig. 10, we believe that quantum effects are not dominant for the described device at temperatures around 300 K, but can be important at lower temperatures.

We further added a new Supplementary Note 3, where we presented 3D simulations. Even though the 3D results have shown some new features, 1D simulations are still qualitatively similar to time-consuming 3D simulations; a new Supplementary Fig. 11 is added to support the claim.

Supplementary Figure 10 | The distribution of the delay time in quantum regime. Red: $\frac{\hbar\omega}{2k_B T}=0.6$; blue: $\frac{\hbar\omega}{2k_B T}=1$; magenta: $\frac{\hbar\omega}{2k_B T}=1.5$; brown: $\frac{\hbar\omega}{2k_B T}=2$. Note that quantum fluctuations result in the maximum of the distributions shifting to the right (we used $U_0/(\hbar\omega/2) = 6$). The probability density is normalized by $\sqrt{\frac{\pi\hbar}{m\omega}}$.

Supplementary Figure 11 | 3D simulations of a diffusive memristor dynamics. (a) Conductance (red curve) normalized by its maximum value and voltage across the memristor normalized by its threshold V_{th} , potentials used in simulations are shown in the inset (see Supplementary Note 3), simulations are done for 9 Ag-nanoparticles. (b) The histogram is fitted by the analytically derived distribution, the fitting is quite good despite of the complex 3D potential, simulations are done for 6 Ag nanoparticles. (c) The possible current paths between memristor electrodes via Ag nanoparticles (for simplicity the case of only 4 Ag nanoparticles are shown). Panels 1-6 show the Ag particles positions at the points marked by orange circles in panel a), panels 1-3 demonstrate how the 3D conducting path are forms, while panels 4-6 show the relaxation of the device to its off state.

Response to Reviewer #3:

The work presents a new solution for random number generation with a diffusive memristor device, consisting of a nanoionic switching element with unstable low-resistance state. The solution for number generation is new, however it shows some limitations in performance.

RESPONSE:

We thank the reviewer for the positive comment on the novelty of our work. We will address the technical comments on the TRNG performance in the next few paragraphs.

In particular, the solution does not overcome one of the limitation that is mentioned in the introductory part of the manuscript, namely that ‘need for complicated probability tracking and careful tuning of the applied voltage/current’. In fact, to reach the random delay time in a desired time range, one should carefully tune the applied voltage to achieve that mean value of delay time.

RESPONSE:

We thank the review for bringing up this discussion, which is helpful for us to further clarify the advantage of our approach. Our circuit is designed to yield unbiased random numbers without probability tracking or any other parameter tuning. Unlike previous proposed memristive TRNG designs, we do not use the median value of any switching characteristic as a threshold to get ‘1’s and ‘0’s. Threshold-based approaches fail when the underlying distribution shifts over time, as occurs with all known stochastic memristors. Our approach, in contrast, should produce truly random numbers no matter what the behavior of the distribution, provided that the clock frequency is sufficiently fast, which is usually not a big problem.

In particular, it can be shown that the Shannon entropy of each generated bit is at least $1 - O\left(\left(\frac{1}{f_{clock}}\right)^2\right)$ for any continuous distribution. This is confirmed by our experimental results, which were produced with **no probability tracking, no post-processing, and no complicated tuning of the applied voltage**. We collected 54 M binary bits from a single diffusive memristor in a continuous run **with all circuit parameters held constant**. These 54 M bits passed the NIST tests, as shown in Supplementary Table 2.

It should be noted that for data collection, an input voltage of 0.5 V was chosen to keep delay time reasonable for the maximum 11.0592 MHz crystal oscillation frequency in the microcontroller and also to minimize power consumption and extend device lifetime, **but other voltages can be used without issue** with on chip integration and faster clock signal.

To compare the advance and novelty of our method with previous efforts, we have added one paragraph into the main text (page 19, paragraph 2). Please refer to the **response 3 to reviewer 1** (pages 4-5 of this response letter) for the added text.

Also, the frequency operation of such TRNG is limited by the relaxation time (e.g. 0.1 ms in Fig. 1c), since it is necessary to wait for the relaxation of the Ag particles after every switching process.

RESPONSE:

We thank the reviewer for pointing out the frequency issue. This is essentially the same concern as reviewer 2 (comment 3, page 10 of this response letter).

For the demo of our proposed diffusive memristor TRNG, we are not using an integrated circuit, which makes the switching speed we can deal with relative slow due to large parasitic effects (capacitances, for example). It is easier and also quicker for us to deal with a slow switching speed with the set-up on bread-boards as shown in the paper. Regarding the switching speed, device engineering can be used and have been successfully demonstrated to significantly short the relaxation time to hundreds of ns.^{2,3} In addition, our proposed method to build a TRNG can be easily applied to other threshold-switching devices. For instance, NbO₂ could be a good candidate because of its fast switching speed (delay time: 0.7 ns and relaxation time: 2.3 ns)⁴. With this speed, the potential bitrate can reach more than 300 MHz even using 1-bit counter. But surely, for faster devices, much faster clock signal is also required, which is not going to be a problem (a stable clock rate of 3 GHz has been used for the CPU (Intel Pentium 4 model) 15 years ago). Therefore, the output rate can be much faster if we adopt a fully integrated circuit.

Moreover, even our current diffusive memristor TRNG (6 kb/s) already shows promises for applications in low-power and low-speed encryption systems, such as internet secure session link (SSL) key, car key and identification cards^{5,6}, as discussed on page 19 in the main text. For high-frequency applications, one traditional approach is to combine such low-frequency TRNG with a linear-feedback shift register^{5,7}, which can greatly increase the bitrate as we have demonstrated in Supplementary Note 4.

The claim that this solution is inherently radiation hard is valid for all memristor-based number generators, not only the present one.

RESPONSE:

We agree with the reviewer that the radiation hard advantage is valid for all memristor-based random number generators. Under the context, we were comparing the memristor based TRNG with other charge transport based ones. To clear the confusion, we have modified the sentence on page 20, paragraph 2 slightly and it now reads:

“Our diffusive memristor TRNG (like all other memristive switching based TRNG) could be more resistant to harsh environments than other electron-based TRNGs.”

Numerical simulations are added to support the mechanism, but the model and explanation are basically the same as in previous works by the same authors [34, 35].

RESPONSE:

First of all, we'd like to clarify that the devices used in the current work are slightly different from devices used in our previous work [34, 35] in terms of purpose, structure and property. Moreover, the goal of previous works [34, 35] were not to analyze the statistics of diffusive memristor switching, so here we have presented the detailed statistical comparison of the

experimental data and simulations for the first time.

In addition, we further develop the 3D model in this work, the results of simulations are in Supplementary Note 3 and Supplementary Fig.11. These are included in this response letter in **response 4 to reviewer 2** (pages 11-13 of this letter). Even though the 3D model provides some further understanding of Ag nanoparticle dynamics, we believe that the 1D model used here is sufficient to understand most statistical properties of two terminal diffusive memristors. The 3D models might become critically important if an additional electric field along y or z axis is applied as is the case for multi-terminal devices.

Finally, we would like to point out that the modeling work is used to here ONLY to support the mechanism as correctly pointed out by the reviewer. The focus of the manuscript is to build a new TRNG using a NOVEL approach rather than to present a new model or a new diffusive memristor.

References

1. Wang, Z., et al. Memristors with diffusive dynamics as synaptic emulators for neuromorphic computing. *Nature Mater.* 16, 101-108 (2017).
2. Midya, R., et al. Anatomy of Ag/Hafnia-based selectors with 10^{10} nonlinearity. *Adv. Mater.* **29**, 1604457 (2017).
3. Song, J., et al. Monolithic integration of AgTe/TiO₂ based threshold switching device with TiN liner for steep slope field-effect transistors. *IEEE Electron. Dev. Meet.* 2016.
4. Pickett, M. D., Williams, R. S. Sub-100 fJ and sub-nanosecond thermally driven threshold switching in niobium oxide crosspoint nanodevices. *Nanotechnology* 23, 215202 (2012).
5. Huang, C. Y., Shen, W. C., Tseng, Y. H., King, Y. C., Lin, C. J. A contact-resistive random-access-memory-based true random number generator. *IEEE Electron Device Letters* **33**, 1108-1110 (2012).
6. Liu, N., Pinckney, N., Hansen, S., Sylvester, D., Blaauw, D. A TRNG using time-dependent dielectric breakdown. *Symp. VLSI Circuits Dig. Tech., Papers*, pp. 216-217, 2011.
7. Fujita, S., et al. Si nanodevices for random number generating circuits for cryptographic security. *In Proc. ISSCC*, pp. 294-295, 2004.

Reviewer #1 (Remarks to the Author):

The authors have fully addressed my questions. Thus I recommend publication.

Reviewer #2 (Remarks to the Author):

Most of questions are well answered by the response letter and revised manuscript, however there're couple questions still remained which need to be clarified further,

1. In terms of temperature effect of bitrate, the authors' explanation and the revised manuscript is as below,

"We collected 11 M binary bits under 1 kHz pulses (voltage amplitude: 0.5 V and pulse width: 300 μ s.). Unlike in operation at room temperature, we can only collect the 3 lowest-order bits, which means the bitrate decreases from 6 to 3 k/s. at 85 oC."

the data trend of bitrate at higher temperatures is different from other publications of RRAM TRNG, can the author calculate the activation energy of noise bitrates based on a wider temperature range and explain why the activation energy is negative in the noise model?

2. Regarding the frequency of TRNG, some applications may not need high data rate, but they still need an uniform data output of the TRNG. What is the longest quiet, no noise data, duration of the TRNG when the authors were taking the measurement with first or post-9M cycles? Please provide the data and the possible solution in terms of this potential problem.

Reviewer #3 (Remarks to the Author):

The authors state that their results were achieved with 'no probability tracking, no post-processing, and no complicated tuning of the applied voltage,' and that 'an input voltage of 0.5 V was chosen to keep delay time reasonable'. This shows that there is a need to tune the voltage within a certain window, which cannot be done manually as in the experiment, but requires a dedicated circuit for probability tracking. If the input voltage is too low, the device cannot switch within the designed pulse-width, while if the input voltage is too high, the device will switch immediately within an early fraction (e.g., 1%) of the overall pulse-width. In both cases, the TRNG will fail. How is this voltage tuned for ideal TRNG operation? Although this might be less complicated than tuning the input voltage to yield 50% probability of switching, it requires a dedicated tuning circuit which should be addressed by the authors and clearly specified for fair comparison of this solution to the state of the art.

Reviewer #1 (Remarks to the Author):

The authors have fully addressed my questions. Thus I recommend publication.

RESPONSE: We thank the reviewer for the positive comment and recommendation.

Reviewer #2 (Remarks to the Author):

Most of questions are well answered by the response letter and revised manuscript, however there're couple questions still remained which need to be clarified further,

1. In terms of temperature effect of bitrate, the authors' explanation and the revised manuscript is as below, “We collected 11 M binary bits under 1 kHz pulses (voltage amplitude: 0.5 V and pulse width: 300 μ s.). Unlike in operation at room temperature, we can only collect the 3 lowest-order bits, which means the bitrate decreases from 6 to 3 k/s. at 85 C.” the data trend of bitrate at higher temperatures is different from other publications of RRAM TRNG, can the author calculate the activation energy of noise bitrates based on a wider temperature range and explain why the activation energy is negative in the noise model?

RESPONSE: Our bitrate is given by switching cycles per second (the frequency of applied pulses to the device) \times collected bits per switching cycle. The frequency of the applied pulses was kept constant at 1 kHz for all temperatures. However, as the temperature increases, the delay time becomes shorter (Supplementary Fig. 6). As a result, the clock frequency needs to be faster at a higher temperature to keep bitrate the same or higher. However, we used the same clock frequency (11.0592 MHz, highest available in the microcontroller) at all temperatures. This explains why at room temperature we were able to collect the 6 lowest-order-order bits while only the 3 lowest-order bits at 85 °C and hence the overall bitrate decreased. The reviewer is right that the activation energy should be positive since the physical switching process happens faster at high temperatures, as shown in Supplementary Fig. 7. In summary, the bitrate decreases at higher temperature because of the capability of the peripheral circuitry, and the temperature dependent device behavior is consistent with other RRAMs used for TRNGs.

2. Regarding the frequency of TRNG, some applications may not need high data rate, but they still need an uniform data output of the TRNG. What is the longest quiet, no noise data, duration of the TRNG when the authors were taking the measurement with first or post-9M cycles? Please provide the data and the possible solution in terms of this potential problem.

RESPONSE: We thank the reviewer for pointing this out. For the random bits collection, we left our TRNG continuously running until permanent device failure. We did not see periods of nonrandom data except in the case of devices stuck ON or OFF. The longest device endurance observed so far is 9 M switching cycles over 2.5 hours of continuous operation. However, many real applications do not require continuous operation. Taking internet secure session link (SSL) key as an example, a typical SSL key has 256 bits. Assuming a new secure internet session every 5 minutes as in Ref.1 in this letter, a single diffusive memristor would be sufficient for about 2 years of SSL key generation. NIST test results for the 54 M bits produced by these 9 M cycles (6 bits per cycle) are shown in Supplementary Table 3. The test results show conclusively that the randomness of our system is consistent throughout the lifetime of a single diffusive memristor.

Device engineering could extend the number of switching cycles (endurance). Other approaches to increase the number of generated bits include but not limited to: 1) increasing the clock frequency; 2) using a LFSR (Supplementary Figure 12); and 3) using an array of devices so that the system can switch to a new device each time a failure is detected.

Reviewer #3 (Remarks to the Author):

The authors state that their results were achieved with ‘no probability tracking, no post-processing, and no complicated tuning of the applied voltage,’ and that ‘an input voltage of 0.5 V was chosen to keep delay time reasonable’. This shows that there is a need to tune the voltage within a certain window, which cannot be done manually as in the experiment, but requires a dedicated circuit for probability tracking. If the input voltage is too low, the device cannot switch within the designed pulse-width, while if the input voltage is too high, the device will switch immediately within an early fraction (e.g., 1%) of the overall pulse-width. In both cases, the TRNG will fail. How is this voltage tuned for ideal TRNG operation? Although this might be less complicated than tuning the input voltage to yield 50% probability of switching, it requires a dedicated tuning circuit which should be addressed by the authors and clearly specified for fair comparison of this solution to the state of the art.

RESPONSE: We thank the reviewer for the question. It is true that 0.5 V was chosen manually, based on measurements of the pulse-response distribution of many devices. The Ag:SiO₂ memristors show acceptable device-to-device variations even when they are fabricated at a university-level-cleanroom. For example, we collected totally 76 M bits from 3 devices with the same 0.5 V pulses. We believe that a fixed 0.5 V input will work reliably for direct use of our diffusive memristor TRNG. In other words, we did not use any dedicated tuning circuit in our TRNG, but designed it based on the known properties of the device. This is much like building a circuit with off-the-shelf components with data sheets of devices.

Reference

1. Liu, N., Pinckney, N., Hansen, S., Sylvester, D., Blaauw, D. A TRNG using time dependent dielectric breakdown. Symp. VLSI Circuits Dig. Tech., Papers, pp. 216-217, 2011.